# A High Quality-Factor Optical Modulator with Hybrid Graphene-Dielectric Metasurface Based on the Quasi-Bound States in the Continuum

**DOI:** 10.3390/mi13111945

**Published:** 2022-11-10

**Authors:** Chaolong Li, Hongbo Cheng, Xin Luo, Ziqiang Cheng, Xiang Zhai

**Affiliations:** 1School of Science, East China Jiaotong University, Nanchang 330013, China; 2Science and Technology on Aerospace Chemical Power Laboratory, Hubei Institute of Aerospace Chemotechnology, Xiangyang 441003, China; 3Institute of Optics and Electronics, Chinese Academy of Sciences, Chengdu 610209, China; 4Key Laboratory for Micro-Nano Optoelectronic Devices of Ministry of Education, School of Physics and Electronics, Hunan University, Changsha 410082, China

**Keywords:** bound states in the continuum, metasurface, graphene

## Abstract

In this paper, we combine the dielectric metasurface with monolayer graphene to realize a high quality(Q)-factor quasi-BIC-based optical modulator, and the corresponding modulation performances are investigated by using the finite-difference time-domain (FDTD) method, which can be well fitting by the Fano formula based on the temporal couple-mode theory. The results demonstrate that bound states in the continuum (BIC) will turn into the quasi-BIC with high Q-factor by breaking the symmetry of every unit of the metasurface. Meanwhile, the amplitude and bandwidth of transmission based on the quasi-BIC mode can be efficiently adjusted by changing the Fermi energy (*E_F_*) of monolayer graphene, and the maximum difference in transmission up to 0.92 is achieved. Moreover, we also discuss the influence of the asymmetry degree to further investigate the modulation effect of graphene on the quasi-BIC mode.

## 1. Introduction

Generally speaking, metamaterials can couple to the incident electromagnetic fields due to their periodic subwavelength structural elements and exhibit properties that do not exist in nature. Thus, they have attracted wide attention and produced pioneering electromagnetic and photonic phenomena in the past two decades [1,2,3,4,5]. However, the strong dispersion and difficulty in fabricating nanoscale 3D structures, especially as high losses due to the use of metallic structures have hampered the wide application of metamaterials. Fortunately, metasurfaces which can be conveniently produced by using lithography and nanoprinting methods, are composed of single or few-layer stacks of planar structures, and suppress the unwanted losses due to their ultrathin thickness in the direction of wave propagation [6]. Moreover, metasurfaces enable a spatially varying optical response, and can modify the properties (e.g., phase, amplitude, propagation direction, and polarization) of electromagnetic waves at will. Many kinds of modes can be excited and observed in the different metasurfaces, such as trapped modes [7,8], guided modes [9,10], toroidal modes [11,12], and the supercavity modes [13,14] based on the physics of bound states in the continuum (BICs), which provides a new platform to design advanced optical devices.

BICs are states that keep localized even if they exist together with a continuum of radiating waves in the same spectrum that can take energy away. As quite general wave phenomena, since BICs were first theorized in quantum mechanics [15], they have been observed in many classic systems, such as water waves and acoustic waves [16,17,18]. Recently, many researchers have paid close attention to the photonic structures for the study of BICs due to their flexibility and tunability, which are hard for the quantum systems. Since D. C. Marinica et al. in 2008 [19] reported a BIC in the photonic structure, a great number of achievements has been obtained for the BICs based on the photonic structure, including the anisotropic-induced BIC [20], the influence of substrate on BIC [21], the topological characteristics [22], and the supercavity modes [14]. More importantly, a quasi-BIC mode usually has a ultra-high quality (Q)-factor and strong electromagnetic field enhancement [23], which has played an important role in many fields, such as optical communication [24], vortex beam control [25], laser [26], optical filter [27], and senor [28]. Thus, the BICs are usually converted into ultra-high Q-factor quasi-BIC modes by introducing perturbations into these systems in application.

In 2018, Y. Xu et al. [29] and Y. Kivshar et al. [30] successively realized BICs into the metasurface, which provides a new approach for the design of optical devices in the subwavelength structure. However, the modulation of the BICs is very hard since the structure parameters of the systems must be cautiously re-designed. However, we can take advantage of the tunability of graphene and combine it with the metasurface to dynamically regulate BICs, which has brought about widespread attention because of its special photoelectric properties. Especially, we can dynamically manipulate the surface conductivity σ of graphene, which is closely related to its Fermi energy *E_F_*, by employing bias voltage [31,32,33]. At present, many different optical devices based on the graphene have been widely reported, such as filters [34,35], absorbers [36,37,38,39], optical senors [11,40], and photodetectors [41,42]. Meanwhile, since graphene behaves like a metal in the mid-infrared and THz regime, which is dominated by the intraband transition of electron, it is frequently used as a special medium to regulate and control the optical characteristics of the plasmonic structures [43,44]. Based on this intuition, we intend to take advantage of monolayer graphene to modulate the quasi-BIC based on metasurface.

In this article, we combine the monolayer graphene with dielectric metasurface to achieve a high Q-factor quasi-BIC-based optical modulator. By using the 3D finite-difference time-domain (FDTD) method, we have ascertained the resonance wavelength of BIC of the dielectric metasurface, and studied the effect of structural asymmetry on BIC. By breaking the symmetry of every unit of the metasurface, BIC will turn into the high Q-factor quasi-BIC. Meanwhile, the amplitude and bandwidth of transmission based on the quasi-BIC resonance can be flexibly adjusted by changing the *E_F_* of monolayer graphene, and the maximum difference in transmission up to 0.92 is realized. Moreover, we also discuss the influence of the asymmetry degree to further investigate the modulation effect of graphene on quasi-BIC mode.

## 2. Structure and Model

As shown in Figure 1, the proposed high Q-factor optical modulator is composed of monolayer graphene and silicon-based arrays of square split-rings on the silica substrate. The thickness (z direction) of square split-rings and silica substrate are *h*_1_ = 200 nm and *h*_2_ = 500 nm, respectively. The periods of square split-rings in the x and y directions are both *P_x_ = P_y_* = 1300 nm. Meanwhile, the lengths of different parts in an unit of arrays of square split-rings are *x*_3_
*= x*_4_ = 100 nm and *y* = 800 nm, respectively. The refractive indexes of silica and silicon are set as 1.44 and 3.45, respectively. The asymmetry parameter α is defined as *α = |x*_1_*−x*_2_*|*/2. Simultaneously, the optical conductivity of graphene is associated with the Fermi energy *E_F_* and carrier mobility *μ* according to the random phase approximation. In the local limit, it is expressed as [45]:(1)$$\sigma (\omega )=\frac{2e^{2}T}{\pi \hbar }\frac{i}{\omega +i\tau^{-1}}\log[2\cosh(\frac{E_{F}}{2K_{B}T})]+\frac{e^{2}}{4\hbar }[H(\omega /2)+\frac{4i\omega }{\pi }\int_{0}^{\infty }d\varepsilon \frac{H(\varepsilon )-H(\omega /2)}{\omega^{2}-4\varepsilon^{2}}]$$
where $H(\varepsilon )=\sinh(\hbar \varepsilon /K_{B}T)/\left[\cosh(E_{F}/K_{B}T)+\cosh(\hbar \varepsilon /K_{B}T)\right].$ Here, the temperature *T* is set as 300 K, the carrier mobility is $\mu =10,000\;{\mathrm{cm}}^{2}/\left(V\cdot s\right)$, and the intrinsic relaxation time is $\tau =\mu E_{F}/e\upsilon_{F}^{2}$, where $\upsilon_{F}\approx c/300$ is the Fermi velocity, and *c* is the speed of light in vacuum. Although this work focuses on numerical simulation research, the proposed structure can be relatively convenient to manufacture experimentally compared with the other optical modulators. The dielectric metasurface based on silica substrate is easy for integration under the current CMOS technology, and monolayer graphene grown by chemical vapor deposition can be transferred over the silicon-based metasurface using standard transfer techniques [46]. The commercial software of Lumerical FDTD Solutions is used for numerical simulation which provides the three dimensions FDTD method. The type of source is plane wave, and the incident light is in the transverse electric polarization. Meanwhile, periodic boundary conditions are applied along the x- and y-directions, and a perfectly matched layer is applied along the z-direction. Unless otherwise specified, we note that a incident light with the electric feld parallel to the x axis illuminates normally to the proposed structure, and *|E*_0_*|* is set to 1. Moreover, it is worth noting that the proposed absorber also demonstrates polarization-insensitivity due to its center symmetric feature. The non-uniform mesh is performed, and the minimum mesh size inside monolayer graphene is set as 0.01 nm and gradually enlarges outside the graphene layer for economizing storage space and decreasing computing time.

## 3. Results and Discussion

Firstly, we consider the transmission characteristics of the proposed metasurface shown in Figure 1a without monolayer graphene. As shown in Figure 2a, when *α =* 0, the bandwidth of resonance is disappearing, and the Q-factor is theoretically infinite. Then, when *α =* 50, a high Q-factor (4719) narrow dip with an asymmetric line shape at 1415.68 nm, that is a quasi-BIC mode appears in the transmission spectrum, which can be well fitting by the Fano formula based on the temporal couple-mode theory [47]:(2)$$T=\frac{t^{2}{\left(\omega -\omega_{0}\right)}^{2}+r^{2}\gamma^{2}\pm 2rt(\omega -\omega_{0})\gamma }{{\left(\omega -\omega_{0}\right)}^{2}+\gamma^{2}}$$
where $\omega_{0}$ is the resonant angular frequency, γ is the radiation rate, and *t* and *r* are the transmission and reflection coefficients with *t*^2^ + *r*^2^ = 1. Moreover, the wavelength of the quasi-BIC mode is decreased and red-shifted with the increase of *α*, as shown in Figure 2b. To illustrate the physical mechanism behind the quasi-BIC in this structure, we will explain by analyzing the electric field distributions. As shown in Figure 2c,d, when the system is broken in symmetry by introducing perturbation, two electric dipole moments excited by the two halves of the square split-rings are not equal; that is, the net dipole moment of the whole structure wil not be zero. Thus, the proposed structure can interchange energy with the continuous free-space radiation mode, and exhibits a magnetic dipole quasi-BIC mode with asymmetric Fano line shape. Moreover, the bandwidth and radiation leakage of the quasi-BIC mode are positively related with *α*, which leads to the decrease of Q-factor of the quasi-BIC mode with the increase of α, as shown in Figure 2b.

To manifest the performance of this optical modulator based on the quasi-BIC, we compare the transmission spectra of the dielectric metasurface without and with different *E_F_* of graphene layer when *α =* 75. As shown in Figure 3, when *E_F_* = 0.8 eV is introduced, the transmission dip is slightly red-shifted. Meanwhile, the dip becomes flat and broadens with the decrease of *E_F_*, which means the Q-factor is also clearly decreased. Thus, the proposed hybrid graphene/silicon/silica metasurface can be used as the efficient optical modulator by changing the *E_F_* of graphene, which can be dynamically manipulated by employing bias voltage.

To explain this phenomenon, we calculate the wavelength dependent conductivity of graphene with different *E_F_* according to Equation (1), which is shown in Figure 4. Clearly, the real conductivity of graphene is always positive when *E_F_* is decreased from 0.8 eV to 0.5 eV. However, the imaginary conductivity of graphene is positive when *E_F_* = 0.8 eV, which leads to the metallic behavior of graphene; the imaginary conductivity of graphene becomes negative when *E_F_* = 0.5 eV, which causes graphene to behave like a lossy dielectric [48]. Moreover, when *E_F_* is decreased from 0.8 eV to 0.5 eV, the real conductivity of graphene grows fast at the resonance wavelength, which means the increase of absorption of graphene [49]. Since the electromagnetic waves re-emitted by the induced quasi-BIC mode of the dielectric metasurface, which are strongly coupled to the absorption of graphene layer [50], the reduced profile of the quasi-BIC mode can ascribe to the increase of absorption of the graphene. Moreover, the Fermi energy of graphene can be modulated by applied bias voltage, which can approximately express as [51] $E_{F}\approx \hbar V_{F}{\left(\frac{\pi \varepsilon_{0}\varepsilon_{Si}V_{g}}{eh_{1}}\right)}^{1/2}$, where *ħ* is the reduced Planck constant *ε*_0_ and *ε_si_* are the permittivity of free space and silicon, respectively, *V_g_* is applied bias voltage, e is the electron charge, and *h*_1_ is the thickness of square split-rings. Thus, the Fermi energy of graphene is increased with bias voltage, and there is a trade-off between the value of bias voltage and the extinction ratio of the proposed modulator. However, the bias voltage required to increase the Fermi energy of graphene to 0.5 eV to 0.8 eV is about 17.393 V, and this power consumption is not large in practical applications. Thus, the proposed modulator can still play an important role in practical applications. Significantly, the real conductivity of graphene increases with the amount of EF of graphene in the THz band, which represents the opposite trend for the near-infrared regime. Significantly, the real conductivity of graphene increases with the amount of *E_F_* of graphene in the THz band, which represents the opposite trend for the near-infrared regime. For this reason, some researchers designed and investigated the graphene-based modulator by setting the high conductivity of graphene in the THz band [52,53].

Then, in order to better investigate the tunability of the proposed modulator, we study the transmission of the structure when *α =* 75 for *E_F_* = 0.8 eV, 0.5 eV, and 0.1 eV, respectively, and calculate the absolute value of the transmission difference $\Delta T_{1}=\left|T(E_{F}=0.8eV)-T(E_{F}=0.1eV)\right|\%$, as shown in Figure 5. It is clear that the maximum of ∆*T*_1_ can reach 58.2% at the resonance wavelength of 1421.31 nm. Thus, an effective and adjustable amplitude modulator can be realized in the near-infrared regime, which can provide potential applications in biosensor and communication.

Next, we will consider the influence of asymmetry parameter *α* to further look into the modulation effect of graphene on the quasi-BIC mode, as shown in Figure 6. When the *α* = 0, that is, there is no perturbation in the structure, the graphene with *E_F_* = 0.5 eV has a slight effect on the transmission spectrum. While the *α* is not equal to 0, that is, in the case of the perturbation, the quasi-BIC mode is excited in the structure, and the same graphene has a huge influence on the transmission spectrum. Furthermore, the max difference of transmission is changed to 0.92 at the wavelength of 1417 nm when *α* = 75. Even if *α* = 50, the max difference of transmission is 0.87 at the wavelength of 1416 nm, which means that a small *α* can achieve good modulation performance for our structure. This also shows its advantage in practical application.

## 4. Conclusions

To conclude, a high Q-factor quasi-BIC-based optical modulator is proposed in the near-infrared regime, and the corresponding modulation performances are investigated by using the FDTD method, which can be well fitting by the Fano formula based on the temporal couple-mode theory. The results demonstrate that BIC will turn into the quasi-BIC with high Q-factor by breaking the symmetry of every unit of the metasurface. Meanwhile, the amplitude and bandwidth of transmission based on the quasi-BIC mode can be flexibly adjusted by changing the *E_F_* of monolayer graphene, and the maximum difference in transmission up to 0.92 is realized. Moreover, we also discuss the influence of the asymmetry degree to further investigate the modulation effect of graphene on the quasi-BIC mode. Thus, we believe our results have potential applications in optical modulators, bio-chemical sensors, and optical filters.

## Figures and Tables

**Figure 1 micromachines-13-01945-f001:**
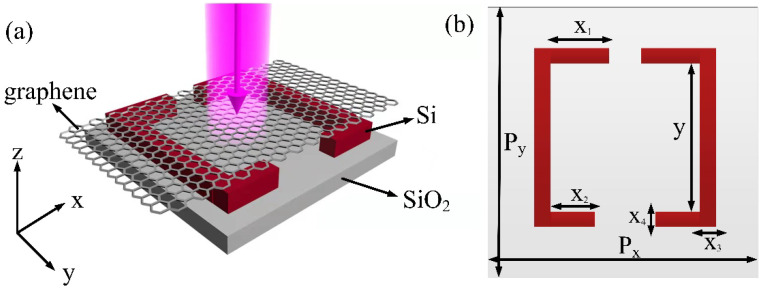
(**a**) Schematic diagram of the hybrid graphene-dielectric metasurface. (**b**) Top view of a unit with dimensions specified.

**Figure 2 micromachines-13-01945-f002:**
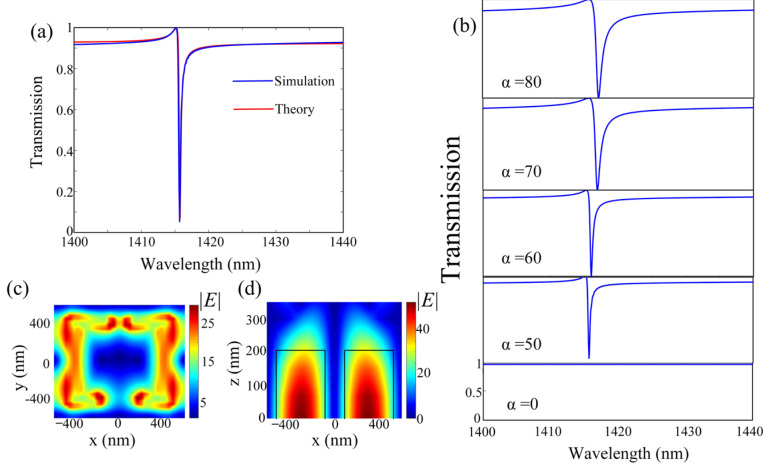
(**a**) The simulated and theoretically fitted transmission spectra of the proposed metasurface with the asymmetry parameter *α* = 50. (**b**) The transmission spectra of the proposed metasurface with different asymmetry parameter α. The distributions of *|E|* of (**c**) top and (**d**) cross section of the structure at the resonance wavelength (λ = 1415.68 nm), respectively.

**Figure 3 micromachines-13-01945-f003:**
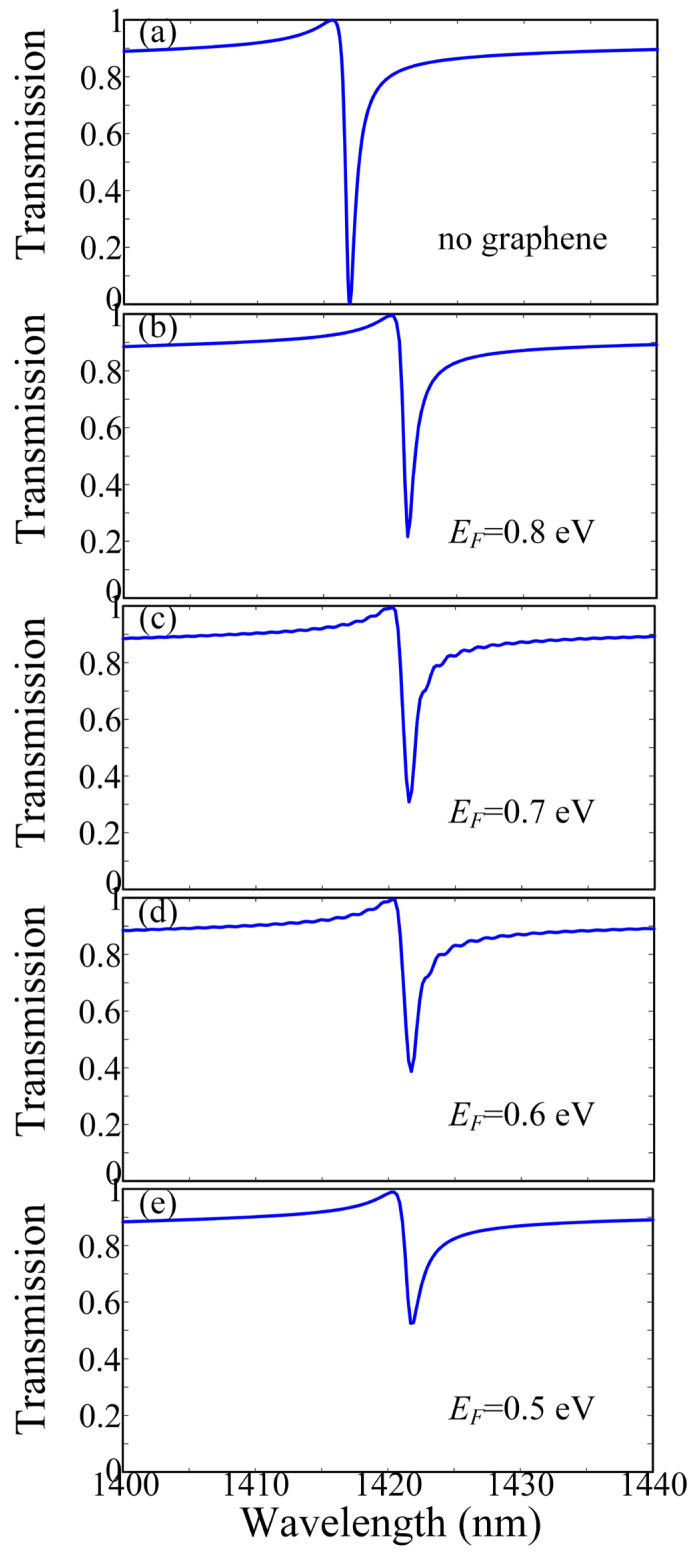
Transmission of the proposed optical modulator (**a**) without and with the different Fermi energies *E_F_* of graphene (**b**–**e**) when α = 75.

**Figure 4 micromachines-13-01945-f004:**
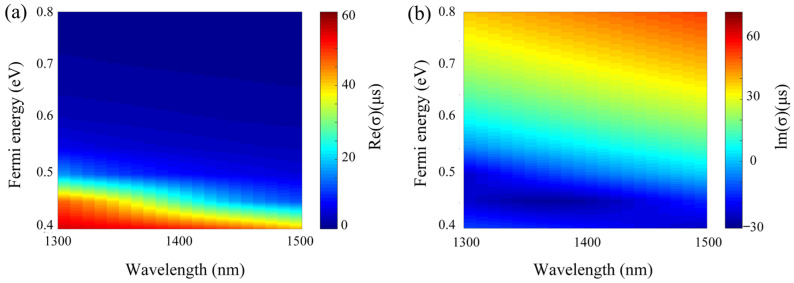
The wavelength dependent (**a**) the real part (**b**) the imaginary part of graphene conductivity with different Fermi energies.

**Figure 5 micromachines-13-01945-f005:**
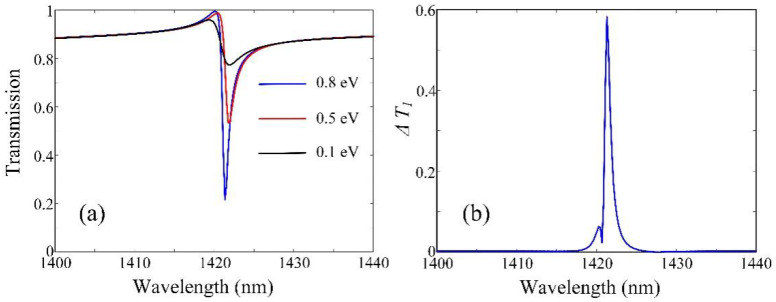
(**a**) Transmission spectra of the proposed modulator with different Fermi energies of the monolayer graphene when *α* = 75. (**b**) The absolute value of the transmission difference between the cases of *E_F_* = 0.8 eV and *E_F_* = 0.1 eV when *α* = 75.

**Figure 6 micromachines-13-01945-f006:**
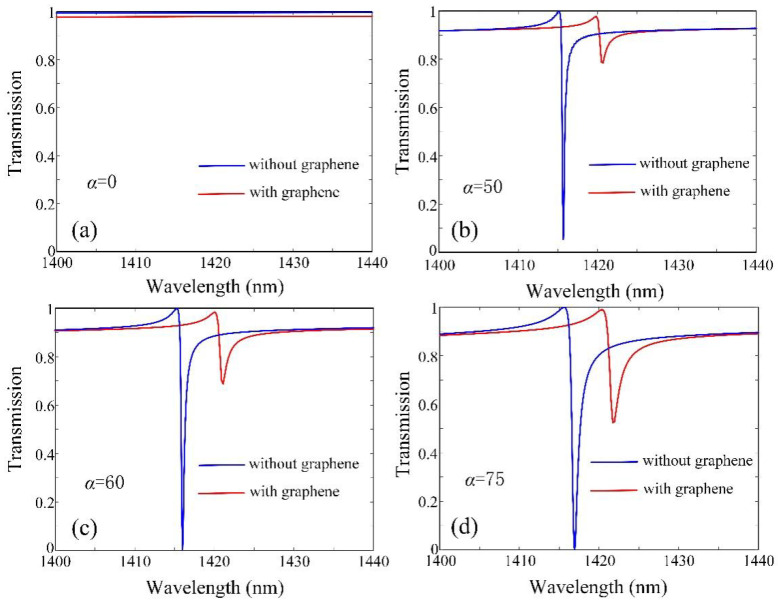
Comparison of the transmission without and with graphene (*E_F_* = 0.5 eV) based on the different (**a**) *α* = 0, (**b**) *α* = 50, (**c**) *α* = 60, (**d**) *α* = 75.

## Data Availability

Data are contained within the article.

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
