# Peer review of "A High Quality-Factor Optical Modulator with Hybrid Graphene-Dielectric Metasurface Based on the Quasi-Bound States in the Continuum"

_micromachines, 2022, doi:10.3390/mi13111945_

Round 1

Reviewer 1 Report

In the manuscript, authors combine the dielectric metasurface with monolayer graphene to realize high quality-factor quasi-BIC based optical modulator, and the corresponding modulation performances are investigated by using the finite-difference time-domain method, which can be well fitting by the Fano formula based on the temporal couple-mode theory. The results demonstrate that bound states in the continuum (BIC) will turn into the quasi-BIC with high Q-factor by breaking the symmetry of every unit of the metasurface. Overall, this work is interesting and supported by adequate data and novelty. I suggest that this manuscript could be accepted with minor revision.

1. In the introduction, authors discuss graphene with a few references. There are two references I recommend which can add in the manuscript.

(1) Y. J. Cai, Z. Y. Wang, S. Yan, L. F. Ye, and J. F. Zhu. Ultraviolet absorption band engineering of graphene by integrated plasmonic structures.Optical Materials Express, 8, 3295-3306 (2018)

(2) Y. J. Cai, Y. B. Guo, Y. G. Zhou, Y. Wang, J. F. Zhu, and C. Y. Chen. Ultracompact and chipless terahertz identification tags using multi-resonant metasurface based on graphene. Journal of Physics D: Applied Physics, 53 015105 (2020).

2. The authors should elaborate on the setup details of the simulation.

3. There are some writting errors. The author should check the manuscript carefully

Author Response

Reviewer #1

Reviewer’s comments:

In the manuscript, authors combine the dielectric metasurface with monolayer graphene to realize high quality-factor quasi-BIC based optical modulator, and the corresponding modulation performances are investigated by using the finite-difference time-domain method, which can be well fitting by the Fano formula based on the temporal couple-mode theory. The results demonstrate that bound states in the continuum (BIC) will turn into the quasi-BIC with high Q-factor by breaking the symmetry of every unit of the metasurface. Overall, this work is interesting and supported by adequate data and novelty. I suggest that this manuscript could be accepted with minor revision.

Comment 1): In the introduction, authors discuss graphene with a few references. There are two references I recommend which can add in the manuscript.

(1) Y. J. Cai, Z. Y. Wang, S. Yan, L. F. Ye, and J. F. Zhu. Ultraviolet absorption band engineering of graphene by integrated plasmonic structures.Optical Materials Express, 8, 3295-3306 (2018)

(2) Y. J. Cai, Y. B. Guo, Y. G. Zhou, Y. Wang, J. F. Zhu, and C. Y. Chen. Ultracompact and chipless terahertz identification tags using multi-resonant metasurface based on graphene. Journal of Physics D: Applied Physics, 53 015105 (2020).

Our reply to comment 1): Thank you for your comment. Related references have been added in the new manuscript.

[34]Y. J. Cai, Z. Y. Wang, S. Yan, L. F. Ye, and J. F. Zhu. Ultraviolet absorption band engineering of graphene by integrated plasmonic structures.Optical Materials Express, 8, 3295-3306 (2018).

[35]Y. J. Cai, Y. B. Guo, Y. G. Zhou, Y. Wang, J. F. Zhu, and C. Y. Chen. Ultracompact and chipless terahertz identification tags using multi-resonant metasurface based on graphene. Journal of Physics D: Applied Physics, 53 015105 (2020).

Comment 2): The authors should elaborate on the setup details of the simulation.

Our reply to comment 2): Thank you for your comment. Related sentences have been added in the new manuscript.

The type of source is plane wave, and the incident light is in the transverse electric polarization. Meanwhile, periodic boundary conditions are applied along the x- and y-directions, and perfectly matched layer is applied along the z-direction. Unless otherwise specified, we note that a incident light with the electric feld parallel to the x axis illuminates normally to the proposed structure and |E0| is set to 1. Moreover, it is worth noting that the proposed absorber also demonstrates polarization-insensitivity due to its center symmetric feature. The non-uniform mesh is performed, and the minimum mesh size inside monolayer graphene is set as 0.01 nm and gradually enlarges outside the graphene layer for economizing storage space and decreasing computing time. (Paragraph 3, Page 2)

Comment 3): There are some writting errors. The author should check the manuscript carefully

Our reply to comment 3): Thank you for your comment. Some sentences have been revised in the new manuscript.

Thus, the BICs are usually convert into ultra-high Q-factor quasi-BICs modes by introducing perturbations into these systems in application. (Paragraph 2, Page 1)

To illustrate the physical mechanism behind quasi-BIC in this structure, we will explain by analyzing the electric field distributions. As shown in Fig. 2(c) and 2(d), when the system is broken symmetry by introducing perturbation, two electric dipole moments excited by the two halves of the square split-rings are not equal, that is, the net dipole moment of the whole structure is no to be zero. (Paragraph 1, Page 4)

Thus, the proposed structure interchanges energy with the continuous free-space radiation mode, and exhibits a magnetic dipole quasi-BIC mode with asymmetric Fano line shape. (Paragraph 1, Page 4)

Significantly, the real conductivity of graphene increases with the amount of EF of graphene in the THz band, which represents the opposite trend for the near-infrared regime. (Paragraph 2, Page 6)

Reviewer 2 Report

The manuscript proposed a high Q-factor quasi-BIC based optical modulator in the near-infrared regime, and the corresponding modulation performances are investigated by using the FDTD method, which can be well fitted by the Fano formula based on the temporal couple-mode theory. The results demonstrate that BIC will turn into the quasi-BIC with high Q-factor by breaking the symmetry of each unit of the metasurface. Meanwhile, the amplitude and bandwidth of transmission based on the quasi-BIC mode can be flexibly adjusted by changing the EF of monolayer graphene, which have potential applications in optical modulators and bio-chemical sensors. Thus, I recommend that this manuscript could be accepted with minor revisions.

1. The abstract should be concise and comprehensive, and the author only needs to write the key content in the abstract.

2.  The serial numbers of equation 1 and equation 2 are incorrect. The authors should carefully check them to avoid such errors

3. What does the letter "c" mean in the Fermi velocity? The author should clarify it celarly.

Author Response

Reviewer #2

Reviewer’s comments:

The manuscript proposed a high Q-factor quasi-BIC based optical modulator in the near-infrared regime, and the corresponding modulation performances are investigated by using the FDTD method, which can be well fitting by the Fano formula based on the temporal couple-mode theory. The results demonstrate that BIC will turn into the quasi-BIC with high Q-factor by breaking the symmetry of every unit of the metasurface. Meanwhile, the amplitude and bandwidth of transmission based on the quasi-BIC mode can be flexibly adjusted by changing the EF of monolayer graphene, which have potential applications in optical modulators and bio-chemical sensors. Thus, I recommend that this manuscript could be accepted with minor revision.

Comment 1): The abstract should be concise and comprehensive, and the author only needs to write the key content in the abstract.

Our reply to comment 1): Thank you for your suggestion. The abstract has been added in the new manuscript.

In this paper, we combine the dielectric metasurface with monolayer graphene to realize high quality(Q)-factor quasi-BIC based optical modulator, and the corresponding modulation performances are investigated by using the finite-difference time-domain (FDTD) method, which can be well fitting by the Fano formula based on the temporal couple-mode theory. The results demonstrate that bound states in the continuum (BIC) will turn into the quasi-BIC with high Q-factor by breaking the symmetry of every unit of the metasurface. Meanwhile, the amplitude and bandwidth of transmission based on the quasi-BIC mode can be efficiently adjusted by changing the Fermi energy (EF) of monolayer graphene, and the maximum difference in transmission up to 0.92 is achieved. Moreover, we also discuss the influence of the asymmetry degree to further investigate the modulation effect of graphene on quasi-BIC mode. . (Abstract, Page 1)

Comment 2): The serial numbers of equation 1 and equation 2 are incorrect. The authors should carefully check them to avoid such errors

Our reply to comment 2): Thank you for your comment. The serial numbers of equation 1 and equation 2 have been revised in the new manuscript. Meanwhile, we have carefully checked the new manuscript.

Comment 3): What does the letter "c" mean in the Fermi velocity? The author should clarify it celarly.

Our reply to comment 3): Thank you for your comment. The letter “c” means the speed of light in vacuum. Related sentence has been added to demonstrate it in the new manuscript.

Here, the temperature T is set as 300 K, the carrier mobility is , and the intrinsic relaxation time is , where is the Fermi velocity, and c is the speed of light in vacuum. (Paragraph 3, Page 2)

Reviewer 3 Report

The author has demonstrated interesting simulation results for a quasi-BIC-based optical modulator with hybrid graphene-Si metasurfaces. The proposed device structure enjoys the advantage of being CMOS technology-compatible and having relatively low optical absorption loss. The author also investigated the effect of the geometry of the metasurface on the transmission performance. I think the manuscript fits the technical scope of the journal. I recommend acceptance of this manuscript with revisions.

I have the following questions/suggestions that I hope the authors can address to improve the manuscript:

  1. In line 131, it seems like a wrong figure cites here. It should be Fig. 2(b) rather than Fig. 3(b).
  2. Please clarify the predicted extinction ratio and required voltage operating with the proposed modulator. How to balance the trade-off between the power consumption and extinction ratio?
  3. How to couple the light out and into the chip effectively?

Author Response

Reviewer #3

Reviewer’s comments:

The author has demonstrated interesting simulation results for a quasi-BIC-based optical modulator with hybrid graphene-Si metasurfaces. The proposed device structure enjoys the advantage of being CMOS technology-compatible and having relatively low optical absorption loss. The author also investigated the effect of the geometry of the metasurface on the transmission performance. I think the manuscript fits the technical scope of the journal. I recommend acceptance of this manuscript with revisions.

I have the following questions/suggestions that I hope the authors can address to improve the manuscript:

Comment 1): In line 131, it seems like a wrong figure cites here. It should be Fig. 2(b) rather than Fig. 3(b).

Our reply to comment 1): Thank you for your suggestion. We are sorry for this error. Related sentence has been revised in the new manuscript.

Moreover, the bandwidth and radiation leakage of the quasi-BIC mode are positively related with α, which leads to the decrease of Q-factor of quasi-BIC mode with the increase of α, as shown in Fig. 2(b). (Paragraph 1, Page 4)

Comment 2): Please clarify the predicted extinction ratio and required voltage operating with the proposed modulator. How to balance the trade-off between the power consumption and extinction ratio?

Our reply to comment 2): Thank you for your comment. The Fermi energy of graphene can be modulated by applied bias voltage, which can approximately express as [54] , where ħ is the reduced Planck constant, ε0 and εsi are the permittivity of free space and silicon, respectively, Vg is applied bias voltage, e is the electron charge, and h1 is the thickness of square split-rings. Thus, the Fermi energy of graphene is increased with bias voltage, and there is a trade-off between the value of bias voltage and extinction ratio of proposed modulator. However, the bias voltage required to increase the Fermi energy of graphene to 0.5 eV to 0.8 eV is about 17.393 V, and this power consumption is not large in practical applications. Thus, the proposed modulator can still play an important role in practical applications. Related sentences and reference have been added to clarify this in the new manuscript.

The Fermi energy of graphene can be modulated by applied bias voltage, which can approximately express as [54] , where ħ is the reduced Planck constant, ε0 and εsi are the permittivity of free space and silicon, respectively, Vg is applied bias voltage, e is the electron charge, and h1 is the thickness of square split-rings. Thus, the Fermi energy of graphene is increased with bias voltage, and there is a trade-off between the value of bias voltage and extinction ratio of proposed modulator. However, the bias voltage required to increase the Fermi energy of graphene to 0.5 eV to 0.8 eV is about 17.393 V, and this power consumption is not large in practical applications. Thus, the proposed modulator can still play an important role in practical applications. (Paragraph 2, Page 5)

  • Gao, J. Shu, C. Qiu, and Q. Xu, “Excitation of Plasmonic Waves in Graphene by Guided-Mode Resonances,” ACS Nano 6(9), 7806-7813 (2012).

Comment 3): How to couple the light out and into the chip effectively?

Our reply to comment 3): Thank you for your comment. As we know, light can effectively couple into the chip in three ways: grating waveguide, prism or cross section coupling.

Reviewer 4 Report

The parameter alpha seems to be essential in the numerical analysis to obtain the simulation results in the manuscript. However, there is no description on the parameter. it appears in Fig. 2 suddenly without any explanation. Therefore, the content of the manuscript is not understandable.

Author Response

Reviewer #4

Reviewer’s comments:

The parameter alpha seems to be essential in the numerical analysis to obtain the simulation results in the manuscript. However, there is no description on the parameter. it appears in Fig. 2 suddenly without any explanation. Therefore, the content of the manuscript is not understandable.

Our reply ): Thank you for your comment. We have added the description on the parameter alpha (α) in the caption of Fig. 2.

Fig. 2. (a) The simulated and theoretically fitted transmission spectra of the proposed metasurface with the asymmetry parameter α=50. (b) The transmission spectra of the proposed metasurface with different asymmetry parameter α. The distributions of |E| of (c) top and (d) cross section of the structure at the resonance wavelength (λ = 1415.68 nm), respectively. (Figure 2, Page 3)

Round 2

Reviewer 4 Report

The manuscript can be accepted for publication.